# Encoding edge type information in graphlets

**Mingshan Jia[1]\*, Maité Van Alboom[2], Liesbet Goubert[2], Piet Bracke[2], Bogdan Gabrys[1], Katarzyna Musial[1]**

**1** Complex Adaptive Systems Lab, School of Computer Science, University of Technology Sydney, Sydney, NSW, Australia, **2** Health Psychology Lab, Ghent University, Ghent, Belgium

\* mingshan.jia@student.uts.edu.au

**Data Availability Statement:** The code, proposed approaches, and experiment results presented in the study are available from https://github.com/MingshanJia/explore-local-structure. Readers can

## Abstract

Graph embedding approaches have been attracting increasing attention in recent years mainly due to their universal applicability. They convert network data into a vector space in which the graph structural information and properties are maximumly preserved. Most existing approaches, however, ignore the rich information about interactions between nodes, i.e., edge attribute or edge type. Moreover, the learned embeddings suffer from a lack of explainability, and cannot be used to study the effects of typed structures in edge-attributed networks. In this paper, we introduce a framework to embed edge type information in graphlets and generate a Typed-Edge Graphlets Degree Vector (TyE-GDV). Additionally, we extend two combinatorial approaches, i.e., the colored graphlets and heterogeneous graphlets approaches to edge-attributed networks. Through applying the proposed method to a case study of chronic pain patients, we find that not only the network structure of a patient could indicate his/her perceived pain grade, but also certain social ties, such as those with friends, colleagues, and healthcare professionals, are more crucial in understanding the impact of chronic pain. Further, we demonstrate that in a node classification task, the edge-type encoded graphlets approaches outperform the traditional graphlet degree vector approach by a significant margin, and that TyE-GDV could achieve a competitive performance of the combinatorial approaches while being far more efficient in space requirements.

## 1 Introduction

Abstracting entities and their interactions as nodes and links, networks are a general model for studying complex systems [1]. Real-world complex networks contain not only topological information but also rich information about nodes and links [2]. Many previous works propose to exploit node attributes by jointly embedding them with topological structures, and the enhanced representation has been shown to be powerful for numerous applications, such as node classification [3–5], link prediction [6, 7], anomaly detection [8, 9], and network visualisation [10].

These approaches, however, overlook rich information about interactions between nodes. Edge attribute or edge type information is indispensable when studying many networks. For instance, the label of each edge in a routing network reflects the cost of traffic via that edge and

also contact author Maité Van Alboom (Maite.vanalboom@ugent.be) in order to obtain access to the data.

**Funding:** This work was supported by the Australian Research Council, Grant No. DP190101087: "Dynamics and Control of Complex Social Networks". The data collection was supported by two grants from the Fund for Scientific Research-Flanders, Belgium (Grant No. G020118N awarded to Liesbet Goubert and Piet Bracke and grant No. 11K0421N awarded to Maité Van Alboom).

**Competing interests:** The authors have declared that no competing interests exist.

is used to determine the best possible routing scheme; in a user-object bipartite network, an edge is labelled with the user's rating for the product, based on which effective recommender systems can be built [11]; and in egocentric social networks, labels of edges illustrate different types of social relationships and are essential in analysing individuals' behaviours and characteristics [12].

To address this issue, we propose to incorporate edge type information into graphlets and form a Typed-Edge Graphlets Degree Vector (TyE-GDV) [13]. This is mainly inspired by the classic graphlets approach that generates a graphlet degree vector (GDV) [14]. Each coordinate in GDV has a clear meaning, i.e., representing a particular topological structure. Due to this excellent explainability, graphlets have gained considerable ground in a variety of domains. It is revealed that in molecular networks, proteins performing similar biological functions possess similar local structures depicted by GDV [15]. Graphlets are also used in computer vision and neuroscience, in order to capture the spatial structure of superpixels [16] or to detect structural and functional abnormalities in the brain [17]. Notably, in social science, egocentric graphlets are used to depict the social interaction patterns of individuals [18]. In the proposed TyE-GDV approach, we choose to add an extra dimension of edge type on top of GDV, that is to say, counting each type of edge touched by each graphlet. Therefore, each coordinate in the two-dimensional vector also has a clear meaning—the number of edges of a certain type in a certain graphlet. We also propose an egocentric version of TyE-GDV that is more succinct and space efficient when dealing with egocentric networks.

We then employ the proposed TyE-GDV and the classic graphlets degree vector [15] to evaluate and analyse a collection of egocentric social networks of chronic pain patients. The real-life data is gathered from two chronic pain leagues in Belgium [19]. Each patient creates an egocentric social network with edges denoted by the type of social relationships. The patients are divided into four groups based on their self-perceived pain grades. First, we find that graphlet patterns are indeed helpful in assessing the pain grade—patients with higher pain grades form more star-like structures (3-star graphlets), whereas patients with lower pain grades have more tightly connected structures (3-cliques, 4-chordal-cycles and 4-cliques). Second, the edge-type embedded graphlets depicted by TyE-GDV provide us with more insights into how particular social ties could affect the perceived pain. Specifically, we find that in patients of higher pain grades, friends and healthcare workers are the dominant social types in the poorly connected 3-stars; and that in patients of lower pain grades, friends and colleagues appear more often in the tightly connected graphlets such as 3-cliques and 4-cliques.

To compare with the proposed method, we further extend two recent graphlets-based approaches, i.e., the colored graphlets approach [20] and the heterogeneous graphlets approach [21], to edge-attributed networks and egocentric networks. We then apply TyE-GDV and the extended colored and heterogeneous graphlets approaches to a node classification task. Besides the egocentric social networks of chronic pain patients, the dataset also contains rich information about the patients' demographic attributes, pain scores and other physical/psychological well-being descriptors, which are used as baseline features in the experiment. We then set up to include features captured by the proposed method and other related approaches and aim to classify patients into different pain grade groups. The result shows that the edge-type augmented graphlet features are more distinctive than the traditional non-typed graphlet features provided by GDV in separating patients with different pain grades.

To summarise, the main contributions of this work are as follows:

- In order to effectively encode edge type information, we propose a novel framework to generate a Typed-Edge Graphlet Degree Vector;

- We further modify the TyE-GDV framework so that it is better suited for egocentric networks;

- We extend colored graphlets and heterogeneous graphlets approaches for edge-typed networks and egocentric networks.

- According to a case study on individuals with chronic pain, certain social ties are more crucial in understanding the effects of chronic pain and may result in more successful therapeutic interventions.

- We demonstrate that rich structural information enhanced by edge-type information leads to significant improvement in a typical machine learning task.

The remainder of this paper is organised as follows. Related works are discussed in Section 2. Preliminary knowledge is provided in Section 3. The proposed typed-edge graphlets, and the extended colored graphlets and heterogeneous graphlets are introduced in Section 4 and Section 5, respectively. Experiments, results and analysis are presented in Section 6. Finally, we conclude and discuss future directions in Section 7.

## 2 Related work

Compared to abundant approaches that take advantage of node attributes, fewer works have focused on leveraging edge attribute information in graph analysis. A straightforward approach is to construct an adjacency matrix containing edge attributes and then to factorise it [22]. This approach, however, involves the expensive matrix operation like the singular value decomposition and therefore lacks scalability. EdgeCentric focuses on the problem of anomaly detection and proposes to aggregate attribute values of edges incident to each node and defines an abnormality scoring function [23]. One limitation of EdgeCentric is that its topological scope is restricted within directly connected edges. The framework GERI proposes to first construct a heterogeneous graph by adding extra bridge nodes that represent node/edge attributes, then take a random walk to sample a node's neighbourhood, and learn its embedding [24]. However, converting attribute information into structural information will also make the attribute information lose its original meaning. Based on the approach of Poincaré embeddings [25], Chen and Quirk recently proposed an embedding method that simultaneously preserves the hierarchical property and edge attributes [26]. This approach is apparently limited in its exclusive focus on hierarchical relationships.

Although these approaches are shown to be effective in some downstream tasks, a common issue about them is that their learned embeddings lack explainability—we do not know what each element of the embedding vector means. They are, therefore, unable to reveal the deeper and, ideally, more easily explainable relationship between a local network structure and an edge attribute.

## 3 Preliminaries

In this section, we introduce the notions of graphlets and orbits, and discuss how they can be adapted in egocentric networks.

### 3.1 Graphlets and orbits

Node degree, being the most basic structural feature, counts the number of edges incident to a node. Graphlets or graphlets degree generalises the idea of node degree by counting the number of graphlets the node participates. Specifically, graphlets are a set of "small connected nonisomorphic induced subgraphs" [14]. Small is to say the size of subgraphs is small, usually no

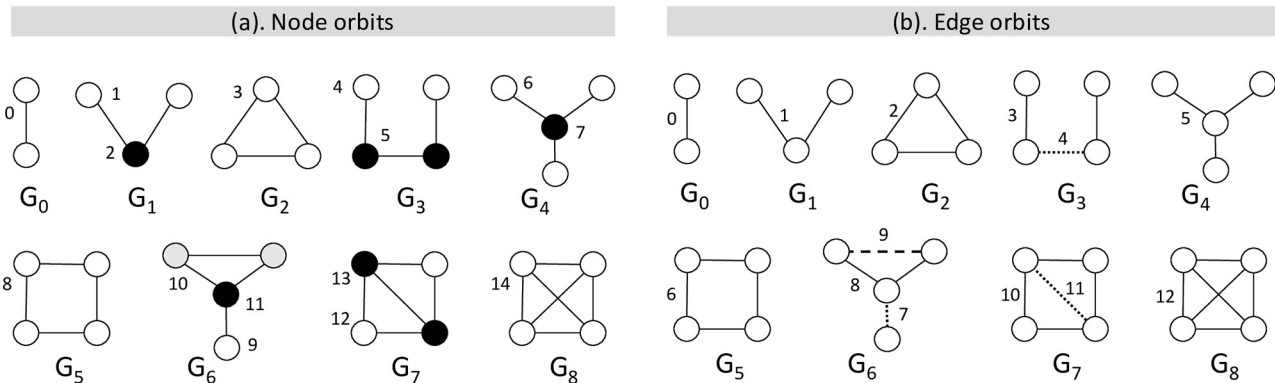

**Fig 1. Graphlets of 2–4 nodes with the enumeration of orbits.** (a) Node orbits: there are in total 15 node orbits, different node colors indicating nonisomorphic node positions within a given graphlet. (b) Edge orbits: there are in total 13 edge orbits, different line types denoting nonisomorphic edge positions within a given graphlet.

more than 4 or 5 nodes. Nonisomorphic means that those subgraphs are structurally distinct, and induced means that all the edges among the nodes in a subgraph need to be considered. The original work covers graphlets of sizes ranging from 2 to 5 nodes, resulting in a total number of 30 different graphlets. Besides, as a node-level structural measure, the non-symmetry of node position is also taken into account, leading to a total number of 73 different subgraph structures, termed automorphism orbits [15]. Briefly, orbits are graphlets that distinguish the position of a focal node (we use orbits and node-orbit graphlets interchangeably in this work). The Graphlet Degree Vector (GDV) of a particular node is thus defined as a vector of the frequencies of 73 orbits.

GDV, or sometimes normalised GDV, has been widely applied in various domains and has become a standard structural feature when measuring the similarities and differences between nodes [15–17]. We summarise node-orbit graphlets of 2 to 4 nodes in Fig 1(a). Taking one of the black nodes in $G_7$, for example, it touches orbit-0 three times (the degree of the node), orbit-2 once (the open triad), orbit-3 twice (the triangle), and orbit-13 once. Therefore, its graphlet degree vector has 3 at the $0^{th}$ coordinate, 1s at the $2^{th}$ and $13^{th}$ coordinates, 2 at the $3^{rd}$ coordinate, and 0s at the remaining coordinates.

The notion of orbits was originally established at a node level, distinguishing a node position when counting graphlets. Hočevar and Demšar later proposed to count graphlets at a link level and introduced the notion of edge orbits [27]. Fig 1(b) gives all edge orbits containing 2 to 4 nodes. Apparently, edge orbits are different from node orbits. For example, there is only one edge orbit in graphlet $G_1$, but two node orbits in it. We also refer to edge orbits as edge-orbit graphlets in this work. The concept of heterogeneous graphlets is built upon edge orbits, and we will discuss more about it in Section 5.

## 3.2 Egocentric graphlets

Graphlets is initially proposed for general networks or sociocentric networks. Although sociocentric networks appear to be more comprehensive modellings of complex systems, collecting sociocentric data via survey is also difficult because participants need to be identifiable to the researcher, and this lack of anonymity can result in unwillingness to participate or bias in responses [12]. Moreover, there are situations where we care more about individuals and their immediate environment. For example, we may want to understand why some people form densely connected ego networks while others don't.

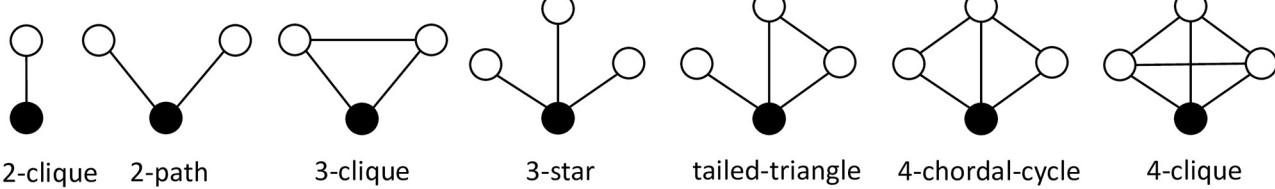

**Fig 2. Egocentric graphlets of 2 to 4 nodes.** There are in total seven egocentric graphlets. The black node in a given graphlet is the ego node, other nodes are alter nodes.

Being a node-level measure, graphlets are naturally suitable to be applied in egocentric networks, with two more restrictions. First, some graphlets that do not fit the definition of an egocentric network need to be eliminated. For example, in graphlets of 2 to 4 nodes (Fig 1(a)), $G_3$ (3-path) and $G_5$ (4-cycle) are excluded because any node in them acting as an ego cannot reach other nodes in a single hop. Second, since only one node can serve as the ego in an egocentric graphlet, it is unnecessary to discriminate between different orbits. Therefore, there are in total seven egocentric graphlets of 2 to 4 nodes, which are 2-clique, 2-path, 3-clique, 3-star, tailed-triangle, 4-chordal-cycle and 4-clique (Fig 2).

## 4 Typed-edge graphlet degree vector

This section describes the framework for generating the typed-edge graphlet degree vector.

The classic graphlet degree vector manages to capture the structural patterns in homogeneous networks. However, many real-world networks also contain rich information on nodes and edges, making them node-attributed, edge-attributed or heterogeneous networks. Information about edge type is particularly important in social networks since it provides a detailed description of relationships among individuals. In the target dataset of this study, for instance, each patient with chronic pain specifies their egocentric social network, including up to ten actors, and each ego-to-alter edge is labelled with one of 13 different types of social ties. In order to analyse edge-attributed networks at a finer granularity and capture the rich edge-typed connectivity patterns, we propose to embed edge type information in graphlets. The original graphlet degree vector generates a one-dimensional vector by counting the instances of each type of graphlet. Here, we propose to build a two-dimensional vector by adding an extra dimension of edge type on top of GDV, that is to say, counting each type of edge contained in each type of graphlet.

We start by formally defining an edge-attributed network.

**Definition 1** *An edge-attributed network G is a triple $\langle V, E, \mathcal{T}_e \rangle$, where $V = \{v_1, v_2, \ldots, v_n\}$ is the set of nodes, $E = \{e_{ij}\} \subset V \times V$ is the set of edges where $e_{ij}$ indicates an edge between nodes $v_i$ and $v_j$, and $\mathcal{T}_e$ is the set of edge types, where $\tau_{e_{ij}}$ denotes the type of edge $e_{ij}$.*

The initial step of the framework is a graph preprocessing, where the set of edge types is mapped to integers ranging from 0 to $|\mathcal{T}_e| - 1$. For example, the 13 different types of social ties in the target dataset are represented from 0 to 12. ($\tau_e \in [0, 12]$). Additionally, the set of orbits $\mathcal{O}$ is converted to integers from 0 to $|\mathcal{O}| - 1$. In this study, we take into account all the node-orbit graphlets within the size of 2 to 4 nodes (Fig 1(a)). Thus, there are 15 orbits coded from 0 to 14 ($o \in [0, 14]$).

**Algorithm 1:** Build Typed-Edge Graphlet Degree Vector.

```
input: preprocessed graph G = ⟨V, E, 𝒯ₑ⟩, set of node-orbits 𝒪, node set
V′.
Output: dictionary dic of vectors for all nodes ∈ V′.
1 initialise: dic = {}
2 foreach i ∈ V′ do
3   initialise a 2d-vector vec of size |𝒪| × |𝒯ₑ| with zeros
```

```
4    foreach o ∈ 𝒪 do
5        Lₑ = GETEDGELIST(o);
6        UPADATE(vec, o, Lₑ)
7    dic[i] = vec;
```

**Algorithm 2:** Update Vector.

```
1  Function UPDATE
   input 2d-vector vec, type of node orbit o, list of edges Lₑ.
2    foreach e ∈ Lₑ do
3        τₑ = GETTYPE(e);
         /* o and τₑ are used as indices in vec.        */
4        vec[o][τₑ] increase by 1;
```

**Algorithm 3:** Code Snippet for Orbit-6, 9 and 10.

```
1  foreach i ∈ V' do
2      initialise a 2d-vector vec of size |𝒪| × |𝒯ₑ| with zeros;
3      foreach u ∈ Nᵢ do
4          foreach v, w ∈ C(Nᵤ, 2) do
             /* oᵢ denotes the corresponding index for orbit-i        */
5          if v ∉ Nᵢ ∧ w ∉ Nᵢ ∧ v ∉ N_w then
6              UPDATE(vec, o₆, [e_{iu}, e_{uv}, e_{uw}]);                     ▷ orbit-6
7          if v ∉ Nᵢ ∧ w ∉ Nᵢ ∧ v ∈ N_w then
8              UPDATE(vec, o₉, [e_{iu}, e_{uv}, e_{uw}, e_{vw}]);             ▷ orbit-9
9          if v ∈ Nᵢ ∧ w ∉ Nᵢ ∧ v ∉ N_w then
10             UPDATE(vec, o₁₀, [e_{iu}, e_{uv}, e_{uw}, e_{iv}]);
11         if v ∉ Nᵢ ∧ w ∈ Nᵢ ∧ v ∉ N_w then
12             UPDATE(vec, o₁₀, [e_{iu}, e_{uv}, e_{uw}, e_{iw}]);           ▷ orbit-10

13
```

Next, for any node of interest, the typed-edge graphlet degree vector (TyE-GDV), i.e., a two-dimensional vector of size $|\mathcal{O}| \times |\mathcal{T}_e|$, is generated using Algorithm 1. Concretely, after initialisation, for each node in a given node set $V'$ and for each orbit in the set of node-orbit graphlets $\mathcal{O}$, the vector is updated through the UPDATE function (Algorithm 2). The calculation of each orbit in Algorithm 1 is omitted for a more concise expression. To demonstrate the detailed process, we give a program snippet for calculating orbit-6, orbit-9 and orbit-10 in Algorithm 3. $C(N_u, 2)$ denotes all possible 2-combinations of the neighbours of node $u$. The use of combinations is to avoid repetitive calculation. In Algorithm 2, $o$ and $\tau_e$ are readily used as indices when updating the vector as a result of the preprocessing stage. Finally, at the end of Algorithm 1, a dictionary of nodes as keys and their corresponding TyE-GDV as values is returned. For example, if an orbit-9 is detected and its four edges are of type '0', '1', '2' and '2', vector elements at coordinates (9, 0), (9, 1), (9, 2) and (9, 2) will increase by 1. Obviously, the time complexity of generating TyE-GDV is the same as counting graphlets. Although the introduction and implementation of the typed-edge graphlets approach is aimed at dealing with edge attributed networks, it can be easily extended to node attributed networks by replacing an edge type with a node type, or to networks containing both different node and edge types by adding an extra dimension of a node type.

As discussed in Section 3.2, egocentric networks are sometimes of special interest, especially when edge type information is included (as in our case study dataset of chronic pain patients). With the restriction of being egocentric, there are fewer orbits in graphlets that need to be considered. Therefore, we propose a tailor-made version of the framework for egocentric networks, called TyE-EGDV (see Algorithm 4). $C(N_i, 2)$ and $C(N_i, 3)$ stand for all possible 2-combinations and 3-combinations of the neighbours of node $i$. Note that in TyE-EGDV, there are in total 7 orbits in $\mathcal{O}$, instead of 15 (see Fig 2). Therefore, the algorithm is more efficient in both time and space.

**Algorithm 4:** Build Typed-Edge Ego-Graphlet Degree Vector.

> **input** : preprocessed graph $G = \langle V, E, \mathcal{T}_e \rangle$, set of egocentric node-orbits $\mathcal{O}$, node set $V'$.
> **output** : dictionary $dic$ of vectors for all nodes $\in V'$.

1 initialise: $dic = \{\}$;
2 **foreach** $i \in V'$ **do**
3 initialise a 2d-vector $vec$ of size $|\mathcal{O}| \times |\mathcal{T}_e|$ with zeros;
4 **foreach** $u \in N_i$ **do**
5 UPDATE($vec, o_0, e_{iu}$); ▷ 2-clique
6 **foreach** $u, v \in C(N_i, 2)$ **do**
7 **if** $v \notin N_u$ **then**
8 UPDATE($vec, o_1, [e_{iu}, e_{iv}]$); ▷ 2-path
9 **else**
10 UPDATE($vec, o_2, [e_{iu}, e_{iv}, e_{uv}]$); ▷ 3-clique
11 **foreach** $u, v, w \in C(N_i, 3)$ **do**
12 **if** $u \notin N_v \wedge u \notin N_w \wedge v \notin N_w$ **then**
13 UPDATE($vec, o_3, [e_{iu}, e_{iv}, e_{iw}]$); ▷ 3-star
14 **else if** $v \in N_u \wedge w \notin N_u \wedge w \notin N_v$ **then**
15 UPDATE($vec, o_4, [e_{iu}, e_{iv}, e_{iw}, e_{uv}]$);
16 **else if** $w \in N_u \wedge v \notin N_u \wedge v \notin N_w$ **then**
17 UPDATE($vec, o_4, [e_{iu}, e_{iv}, e_{iw}, e_{uw}]$); ▷ tailed-tri
18 **else if** $w \in N_v \wedge u \notin N_v \wedge u \notin N_w$ **then**
19 UPDATE($vec, o_4, [e_{iu}, e_{iv}, e_{iw}, e_{vw}]$);
20 **else if** $u \in (N_v \cap N_w) \wedge w \notin N_v$ **then**
21 UPDATE($vec, o_5, [e_{iu}, e_{iv}, e_{iw}, e_{uv}, e_{uw}]$);
22 **else if** $v \in (N_u \cap N_w) \wedge w \notin N_u$ **then**
23 UPDATE($vec, o_5, [e_{iu}, e_{iv}, e_{iw}, e_{uv}, e_{vw}]$); ▷ 4-chord-cyc
24 **else if** $w \in (N_u \cap N_v) \wedge v \notin N_u$ **then**
25 UPDATE($vec, o_5, [e_{iu}, e_{iv}, e_{iw}, e_{uw}, e_{vw}]$);
26 **else**
27 UPDATE($vec, o_6, [e_{iu}, e_{iv}, e_{iw}, e_{uw}, e_{vw}, e_{uv}]$); ▷ 4-clique
28 $dic[i] = vec$;

## 5 Typed-edge degree, colored graphlets and heterogeneous graphlets

Since a node degree is the simplest network structural metric, a naive way of encoding edge type information in a network structure is first to have the notion of a typed-edge degree. Formally, the typed-edge degree of a node $i$ with an edge type $t$, i.e., $d_i^t$, is defined as the number of edges of type $t$ that are connected to $i$. Then, a typed-edge degree vector (TyE-DV) can be defined as a vector containing typed-edge degrees of all types.

Some other approaches that also aim to take a node and/or an edge type into consideration include the colored motifs [28], colored graphlets [20] and heterogeneous graphlets [21]. Colored motifs, as the name suggests, extended G-Tries algorithm that counts motifs [29] by including the information of a node or edge type. This approach, however, is at the network level and is therefore not suitable for a node-level analysis.

Colored graphlets approach [20] is at the node level, and proposes to distinguish different graphlets according to all combinations of node types. The approach is said to be able to deal with typed edges, but without theoretical explanation or experimental demonstration. The article alleges that the total number of combinations equals $2^T - 1$, where $T$ is the total number of possible node types. This is incorrect as it fails to take the size of the graphlet into account. When graphlet size is smaller than the number of node types, the total number of combinations will be smaller than $2^T - 1$. For example, when we consider the graphlet $G_0$, i.e., 2-clique, with

three possible node types, there are in total six combinations, instead of seven. The combination containing all three types cannot exist since there are only two nodes in this graphlet. Below, we give the amended equation for calculating the number of combinations in a given graphlet $g$:

$$\mathcal{C}(g) = \sum_{n=1}^{\min(K(g),T)} \binom{T}{n}, \tag{1}$$

where $K(g)$ is the number of nodes of the graphlet when $T$ refers to a node type, or the number of edges of the graphlet when it refers to an edge type. Note that when $K(g) \geq T$, the equation becomes $\sum_{n=1}^{T} \binom{T}{n}$, which equals $2^T - 1$. We then develop a colored graphlets approach for edge-typed networks, named ColoredE-GDV, which is also applied to the case study in the next section.

The recently proposed heterogeneous graphlets approach [21] also considers a node type in graphlets. It is different from the colored graphlets approach in two ways. First, heterogeneous graphlets are computed at a link level. It distinguishes the position of a given edge, instead of a given node (please refer to the notion of edge-orbit graphlets in Section 3.1). The benefit of a link-based computation is that it is more time-efficient in sparse networks than node-based approaches. The downside, apparently, is that it is not suitable for a node-level analysis. Second, heterogeneous graphlets propose to use combinations with repetitions of node types, rather than just a combination, when distinguishing different graphlets. The total number of possible heterogeneous graphlets is calculated as:

$$\mathcal{H}(g) = \sum_{n=1}^{T} \binom{T}{n} \cdot \binom{K(g) - 1}{n - 1} = \binom{T + K(g) - 1}{K(g)}. \tag{2}$$

Similarly, $K(g)$ is the number of nodes of the graphlet when $T$ refers to a node type, and the number of edges when it refers to an edge type. Since type repetition is allowed in heterogeneous graphlets, the number of possible heterogeneous graphlets is larger than that of colored graphlets.

In order to extend the idea of heterogeneous graphlets to a node-level analysis and to deal with typed edges, we propose a node-based typed-edge heterogeneous graphlets approach, named HeteroE-GDV$^N$ (the original link-based typed-node approach is noted as HeteroN-GDV$^L$). The approach of HeteroE-GDV$^N$ is demonstrated through Algorithm 5. We see clearly that its time complexity stays the same when counting untyped graphlets, but the space complexity grows fast with the number of edge types.

**Algorithm 5:** Node-based Heterogeneous Graphlets Degree Vector (Hetero-GDV$^N$)

**input:** preprocessed graph $G = \langle V, E, \mathcal{T}_e \rangle$, set of node-orbits $\mathcal{O}$, node set $V'$.
**output:** dictionary $dic$ of vectors for all nodes $\in V'$.

```
1  initialise: dic = {};
2  L^{T_e} = [0, 1, ..., |T_e| - 1];
   /* range of edge number of graphlets of size 2−4 nodes        */
3  for k ← 1 to 6 do
4      L_k = [GetCombWithRep (L^{T_e}, k)];
5  foreach i ∈ V' do
6      for o ← 0 to |O| − 1 do
7          initialise vec_o;
8      foreach o ∈ O do
9          k = GetNumOfEdge(o);
10         L_e = GetEdgeList(o);
11         tup = (Sort(L_e));
12         vec_o[GetIndex(L_k, tup)] increase by 1;
13     vec = [vec_0, vec_1, ..., vec_{|O|-1}];
14     dic[i] = vec;
```

**Table 1. Time and space complexities of four approaches that deal with edge type information.** $S$ is the maximum number of nodes in graphlets, K is the maximum number of edges in graphlets, $|\mathcal{O}_e|$ is the number of edge-orbit graphlets.

| Approach | Time complexity | Space complexity |
|---|---|---|
| Colored-GDV [20] | $O(\lvert V\rvert \cdot k_{max}^{S-1})$ | $O(\lvert V\rvert \cdot \lvert\mathcal{O}\rvert \cdot 2^{\lvert T_e\rvert})$ |
| Hetero-GDV$^{L}$ [21] | $O(\lvert E\rvert \cdot k_{max}^{S-2})$ | $O(\lvert E\rvert \cdot \lvert\mathcal{O}_e\rvert \cdot {}^{K}C_{\lvert T_e\rvert+K-1})$ |
| HeteroE-GDV$^{N}$ | $O(\lvert V\rvert \cdot k_{max}^{S-1})$ | $O(\lvert V\rvert \cdot \lvert\mathcal{O}\rvert \cdot {}^{K}C_{\lvert T_e\rvert+K-1})$ |
| TyE-GDV | $O(\lvert V\rvert \cdot k_{max}^{S-1})$ | $O(\lvert V\rvert \cdot \lvert\mathcal{O}\rvert \cdot \lvert\mathcal{T}_e\rvert)$ |

Although the above approaches seem powerful to capture all possible combinations (or combinations of repetitions) of different types of nodes or edges, their numbers of possible graphlets, which are also their space complexities, grow near-exponentially with the number of node or edge types. For example, with 9 node types, in the colored graphlets approach, there are 255 possible colored graphlets for a graphlet of 4 nodes; and in the heterogeneous graphlets approach, there are 495 possible graphlets. In comparison, the space complexity grows linearly with the number of edge types in the proposed TyE-GDV approach. Moreover, out of this large number of possible graphlets, only a tiny percentage of them actually exists in real networks. For example, in Cora citation network [30], only 19 heterogeneous graphlets exist out of 210 possible ones in a 4-clique graphlet.

In order to utilise the colored graphlets and the heterogeneous graphlets approaches in egocentric networks, we further develop their egocentric versions, and apply them in the chronic pain case study. With fewer node orbits to consider, egocentric colored graphlets and egocentric heterogeneous graphlets are faster and more space-saving than the original ones. The implementation of these algorithms is available at https://github.com/MingshanJia/explore-local-structure.

To conclude this section, we summarise the time and space complexities of the four main approaches in Table 1. Colored-GDV, HeteroE-GDV$^{N}$ and TyE-GDV share the same time complexity because they are all node-based algorithms. Hetero-GDV$^{L}$ as the only link-based algorithm, could be faster in sparse networks. When it comes to space complexity, the proposed TyE-GDV grows linearly with the number of edge types, while the other three methods grow near exponentially with it.

## 6 Experiments and analysis

In this section, we apply the proposed methods to analyse the egocentric social networks of chronic pain patients.

### 6.1 Dataset

The real-world dataset is collected from chronic pain patients of the League for Rheumatoid Arthritis, the League for Fibromyalgia and the Flemish Pain League [19]. Each patient creates their own egocentric social networks containing up to 10 alters using the graphical tool GENSI [31]. The types of social ties between the patient (the ego node) and his/her contacts (the alters) are explicitly given. There are in total 13 types of social relationships, including families, friends, colleagues, neighbours, etc. The full list of social ties and their total occurrences are listed in Table 2). The patients were also asked to fill out a questionnaire on pain-related and sociodemographic information. In addition to that, a daily diary consisting of items measuring pain intensity, and physical, psychological and social well-beings, was provided to participants for 14 consecutive days. After eliminating inconsistent and incomplete entries, the final dataset

**Table 2. 13 types of social relationships and their total number of occurrences in 303 egocentric networks.**

| Social Relationship | Type Code | Number of Occurrences |
|---|---|---|
| Partner | T-1 | 222 |
| Father/Mother | T-2 | 209 |
| Brother/Sister | T-3 | 293 |
| Children/Grandchildren | T-4 | 493 |
| Friend | T-5 | 506 |
| Family-in-law | T-6 | 207 |
| Other family | T-7 | 142 |
| Neighbour | T-8 | 69 |
| Colleague | T-9 | 57 |
| Healthcare worker | T-10 | 233 |
| Member of organisations | T-11 | 74 |
| Acquaintance | T-12 | 15 |
| Other | T-13 | 17 |

consists of the egocentric social networks, sociodemographic and pain characteristics of 303 patients. The average age of all patients is 53.5±12 years, including 248 females and 55 males.

Some basic characteristics of the egocentric networks, such as the ego nodes' degree distribution and their edge-type distribution, are shown in Fig 3. The edge-type distribution is computed by summing over all ego nodes on each type of the edges, which is also displayed in the third column of Table 2. The degree distribution reveals that the majority of patients (62%) have ten social connections in their social networks (Fig 3a). However, we do not anticipate node degree to be a discriminative feature in the following analysis since ten contacts are the upper limit in the dataset. According to the edge-type distribution (Fig 3b), the most frequent types in these networks are T-5 "friend" and T-4 "children/grandchildren". In contrast, edge types T-8 "neighbour", T-9 "colleague" and T-11 "member of organisations" are underrepresented. T12 "acquaintance" and T-13 "other" are almost negligible because people would first list their strongest contacts with the limitation of ten connections, leaving little room for those weaker ties.

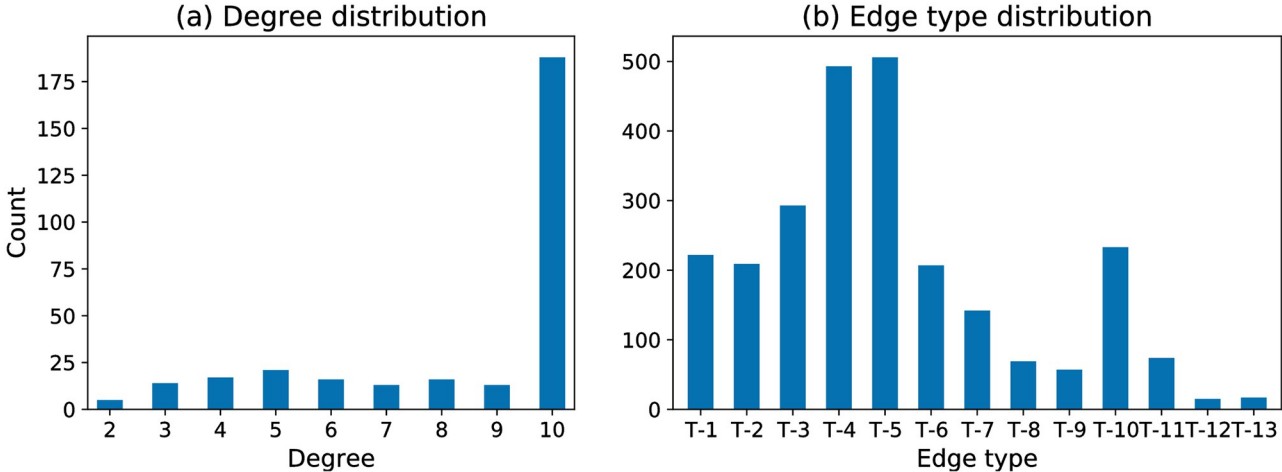

**Fig 3. Degree distribution and edge-type distribution of 303 egocentric social networks.**

Moreover, the grades of chronic pain are calculated by means of the Graded Chronic Pain Scale (GCPS), which evaluates both pain disability and pain intensity [32]. Then, patients are divided into five grades based on their average intensity and disability scores: grade-0 for no pain; grade-1 for low intensity and low disability; grade-2 for high intensity and low disability; grade-3 for moderate disability irrespective of pain intensity; and grade-4 for high disability irrespective of pain intensity. Due to the fact that all participants have a certain degree of chronic pain, their GCPS grades vary from grade-1 to grade-4. Specifically, there are 21 patients in grade-1, 33 patients in grade-2, 67 patients in grade-3 and 182 patients in grade-4. In this study, we aim to investigate whether the structural feature, especially the edge type augmented structural feature captured by TyE-GDV, are helpful in understanding the patients' pain grades.

## 6.2 Analysing pain grades

Evidence within the fields of pain and rehabilitation science has shown that social interactions play an important role in the perception of pain [33]. Perceived social support and pain inference are found to be associated in individuals with chronic musculoskeletal pain [34]. Lower levels of social support and higher levels of pain intensity are observed in rheumatoid arthritis patients at the 3- and 5-year follow-ups [35]. It has also been demonstrated recently that reduced social isolation accounts for significant improvements in self-reported emotional and physical functioning [36]. Typically in these studies, the social milieu of a patient is assessed by the Social Support Satisfaction Scale (ESSS) [37] or the Patient Reported Outcome Measurement Information System (PROMIS®) [38]. However, as these measurements are not based on the real social networks of the patients, they are unable to shed light on the impact of network topologies, especially certain types of interactions, on the perception of pain. To address this issue, we choose to apply both the traditional graphlets approach and the proposed typed-edge graphlets approach to analyse the egocentric networks of chronic pain patients.

First, in order to investigate the impact of network structure on pain grade, we calculate the average egocentric graphlet degree vector for each GCPS grade. A radar chart shows the average values of the seven egocentric graphlets at each grade (Fig 4). We observe clearly that patients with higher pain grades (grade-3 and grade-4) possess more star-like structures (3-star graphlet) in their social networks, whereas patients with lower pain grades (grade-1 and grade-2) compose more clique-like or quasi-clique-like structures (3-clique, 4-clique and 4-chordal-cycle graphlets). A poorer-connected star-like structure denotes a more isolating social setting, whereas a better-connected structure, such as the 3-clique or 4-clique, may suggest stronger social support. These findings are in agreement with the aforementioned studies [33–36] and provide further evidence that a patient's social network may influence how much pain they perceive. Additionally, we discover that the number of immediate connections (2-cliques) is ineffective in differentiating pain grades, which may be partially caused by the limited number of contacts in the dataset. Nevertheless, Evers et al. [35] also discovered that changes in pain are not substantially correlated with the size of a patient's egocentric social network. Jia et al. revealed that the clustering coefficient and the quadrangle coefficient are useful topological features in assessing the perception of pain [39]. These findings further underline the need to consider more complex network topologies when examining patients' social networks.

Furthermore, in order to analyse the association between the types of social ties and the perception of pain, we employ the typed-edge graphlet degree vector and focus on two specific graphlets, namely the weakly connected 3-star graphlet and the highly connected 4-clique graphlet. These two graphlets are selected not only because they represent two extremes of 4-node structures but also because distinct differences between patients with lower pain grades

## Average GDV of different GCPS grades

**Fig 4. Radar chart of average GDV of different GCPS grades.** Each spoke represents the average number of graphlets belonging to that type.

and patients with higher pain grades are observed in them. We first calculate the average counts of the 13 edge types at each pain grade for the 3-star graphlet, i.e., the 3$^{rd}$ row of the Typed-Edge Ego-Graphlet Degree Vector (see Algorithm 4), and draw a parallel coordinates plot (Fig 5(a)). We discover that in the poorly connected star-like structure, edges of type T-5 "friend" and T-10 "healthcare worker" are significantly more frequent in patients with higher pain grades than in patients of lower pain grades. That is to say, in the social networks of higher pain grade patients, friends and healthcare workers are in a rather isolated position—not well connected with other contacts of the patient. Thus, it provides the potential for treatments that boost a patient's friends' and healthcare professionals' social involvement to improve chronic pain management.

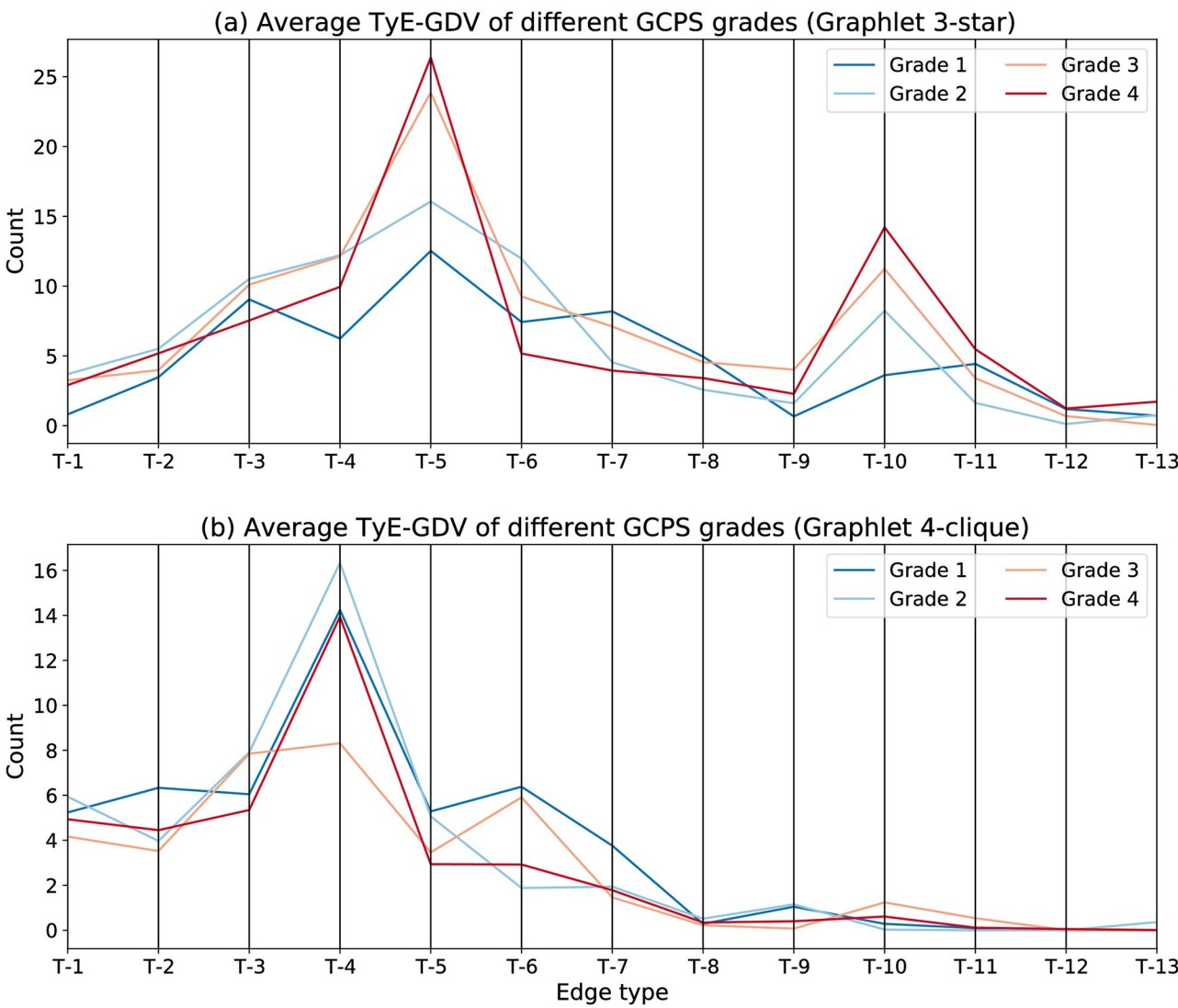

**Fig 5. Two parallel coordinates plots revealing the association of edge type and pain grade.** (a). Average TyE-GDV of four GCPS grades for 3-star graphlet. (b). Average TyE-GDV of four GCPS grades for 4-clique graphlet.

We then calculate the average counts of the 13 edge types at each pain grade for the 4-clique graphlet, i.e., the 6th row of the Typed-Edge Ego-Graphlet Degree Vector, and the corresponding parallel coordinates plot is given in Fig 5(b). We observe that, in this tightly-connected structure, patients with lower pain grades have more edges of type T-5 "friend" than patients with higher pain grades. In other words, friends are better involved in the social networks of patients who perceive lower level pain grades than those who perceive higher pain grades. The importance of friendship is revealed in both 3-star and 4-clique graphlets. As pointed out by other studies [40, 41], people with severe chronic pain may be more liable to a deterioration of their friend relationships and are in more need of supportive behaviours from friends. Another noticeable difference between patients of lower pain grades and patients of higher pain grades is found in edge T-9 "colleague". In contrast to the lower pain grade group, where more than one colleague appears in the clique structures (1.1 on average), colleagues hardly exist in them among the higher pain grade group (0.24 on average). This could be a result of the negative

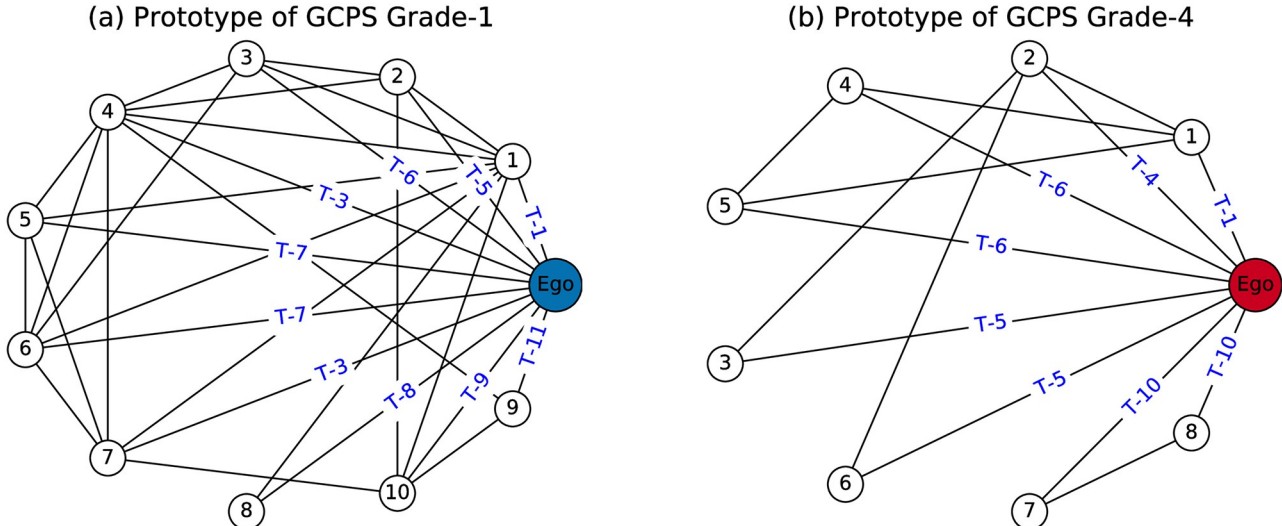

**Fig 6. Social network prototypes of patients with GCPS grade-1 and patients with GCPS grade-4.** (a). In the prototype network of patients with pain grade-1, contacts are tightly connected to each other with the appearance of T-5 friend and T-9 colleague; (b) In the prototype network of patients with pain grade-4, contacts are loosely connected with limited links incident to T-5 friend and T-10 healthcare workers.

consequences that severe chronic pain has on patients' capacity for work [42]. To provide an intuitive grasp of the edge type encoded structural differences between the social networks of patients with different pain grades, we extract two real examples from the dataset as the network prototypes of patients of pain grade-1 and patients of pain grade-4, respectively (Fig 6).

This experiment demonstrates that the extra edge type information encoded in TyE-GDV provides us with more insights into the association between patients' perception of pain grade and the type of social ties in their egocentric networks. It thus has implications for improving therapeutic interventions through boosting particular types of social interactions.

## 6.3 Node classification

We now apply the proposed TyE-GDV, and the extended egocentric versions of colored graphlets (ColoredE-GDV) and heterogeneous graphlets (HeteroE-GDV$^N$) approaches in a typical machine learning task.

Node classification, being one of the most popular and extensively explored tasks in network science [43], aims to predict the labelling of nodes based on a subset of nodes that have ground-truth labels. Here, our goal is to predict the GCPS grade of patients with chronic pain. In order to evaluate the effectiveness of the proposed approaches, we fit six sets of features into a random forest classifier. The first set comprises the patients' demographic attributes, pain-related descriptors and their physical and psychological well-being indicators. Since it contains no network-related information, we refer to it as the raw feature set. The second set and the third set add the typed-edge degree vector (TyE-DV) and the traditional graphlet degree vector (GDV), respectively, on top of the raw features. The fourth set combines the raw features with the proposed typed edge graphlet degree vector (TyE-GDV), and finally, the fifth set and the sixth set plus the colored graphlets degree vector (ColoredE-GDV) and the heterogeneous graphlets degree vector (HeteroE-GDV$^N$), respectively, to the raw feature set.

Since the dataset is not big and the distribution of the four pain grades is not balanced (see Section 6.1), we adopt a stratified 5-fold cross-validation [44] to evaluate the classification

**Table 3. Result table of node classification, reported in the average macro-F1 score (± standard deviation), the average percentage gain over the raw feature set, and the total running time of 500 repetitions.**

|  | Macro F1 (Mean ± Std) | Gain over raw feat. (Mean) | Time in sec. (Sum) |
|---|---|---|---|
| Stratified | 0.248 ± 0.024 | — | 3 |
| Raw feat. | 0.578 ± 0.005 | — | 116 |
| Raw feat. + TyE-DV | 0.600 ± 0.005 | 3.8% | 130 |
| Raw feat. + GDV | 0.597 ± 0.008 | 3.3% | 138 |
| Raw feat. + ColoredE-GDV | 0.608 ± 0.006 | 5.2% | 2091 |
| Raw feat. + HeteroE-GDV[N] | 0.638 ± 0.006 | 10.4% | 8230 |
| Raw feat. + TyE-GDV | 0.619 ± 0.004 | 7.1% | 252 |

performance with different feature sets. Plus, we repeat the above step 500 times and report the mean metric score given the stochastic nature of decision tree-based models.

Table 3 lists the prediction results for six models. As this is a multi-class classification task, and the distribution of the four classes is imbalanced, the macro-F1 score is selected as the evaluation metric. A naive classifier named Stratified is also added to the table (the first row), which simply generates predictions by adhering to the class distribution in the training set. We see clearly that the bottom three approaches that encode type information in graphlets (raw features plus ColoredE-GDV, raw features plus HeteroE-GDV[N], and raw features plus TyE-GDV) perform better than the set of raw features plus TyE-DV and the set of raw features plus GDV. Recall that TyE-DV captures edge type information but with very limited structural information, and GDV, on the other hand, captures the rich structural information but without edge type information. This evidently shows that combining edge type information and rich structural information could lead to more distinctive features in network learning tasks.

We also observe large differences in the running time of those methods. The running time of the set of raw features plus ColoredE-GDV, and especially the set of raw features plus HeteroE-GDV[N] are many times higher than other methods. This is because our dataset has 13 types of edges and the lengths of vectors generated from these two methods grow near exponentially with the number of edge types $|\mathcal{T}_e|$. Correspondingly, the speed of the machine learning algorithm will slow down as the feature vector becomes larger. Table 4 gives the vector lengths of all five approaches. Note that there is no edge type information between alter nodes in many egocentric networks, including this case study dataset. Thus, our implementations of ColoredE-GDV and HeteroE-GDV[N] have excluded all the impossible combinations. Overall speaking, the proposed TyE-GDV is able to achieve a competitive performance while maintaining a small vector length.

## 6.4 Limitations and future directions

Here, we describe some limitations of this work and outline how these might be overcome in future studies.

**Edge direction.** Our current work is limited to undirected networks. To encode edge type information in directed networks, a natural extension of our approach is to apply the notion of directed graphlets [45–47]. The potential approach would be more complex due to the larger

**Table 4. Comparison of vector length of different approaches.**

| Approach | GDV | TyE-DV | TyE-GDV | ColoredE-GDV | HeteroE-GDV |
|---|---|---|---|---|---|
| Len. of vector | 7 | 13 | 91 | 12367 | 38870 |

number of directed node-orbit graphlets. For example, even without considering bidirectional edges, there are in total 40 directed graphlets and 128 directed node orbits for graphlets of 2 to 4 nodes [45].

**Temporal information.** The proposed approach is static or time-independent. To make it suitable for more real-world networks that have nodes and edges appearing and disappearing over time, a potential future work would be studying how to encode edge type or node type information in temporal graphlets [48]. With the extra dimension of time, the potential extension could be beneficial in predicting types of future links or nodes [49, 50].

**Potential applications.** Apart from social networks, the typed edge graphlets approach could be convenient in studying biological networks, especially molecular graphs, where link attributes or bond types are essential information. The proposed approach is promising to be applied in biological network alignment, which aims to find a node mapping between molecular networks that reveals similar network regions [20, 51]. Moreover, inspired by recent works that include subgraph counting in Graph Neural Networks [52, 53], an interesting avenue is to incorporate the edge type enhanced structural information in GNN's message passing scheme.

## 7 Conclusion

In this paper, we propose to encode edge type information in graphlets and introduce the framework for generating the Typed-Edge Graphlets Degree Vector for both sociocentric and egocentric networks. Moreover, we extended the colored graphlets approach and the heterogeneous graphlets approach to edge-typed networks and egocentric networks. Following the application of the traditional graphlet degree vector and the proposed TyE-GDV to the chronic pain patient dataset, we discover that 1) a patient's social network structure could inform their perceived pain; and 2) the extra edge type information encoded in TyE-GDV provides us with more insights into the association between specific social relationships and patients' perception of pain.

We also showed that the rich structural information combined with the edge type information results in a significant improvement of a typical machine learning task that predicts patients' pain grades. Due to the simplicity and excellent explainability, we anticipate that the typed edge graphlets approach would become a standard approach in studying edge-attributed networks and be applied in various tasks.

## Acknowledgments

The authors thank the editors and anonymous reviewers for their excellent comments and suggestions. The authors would also thank Volker Ahlers and Yu-Xuan Qiu for their helpful comments and discussions.

## Author Contributions

**Conceptualization:** Mingshan Jia, Maité Van Alboom, Katarzyna Musial.

**Data curation:** Maité Van Alboom, Liesbet Goubert, Piet Bracke.

**Formal analysis:** Mingshan Jia.

**Funding acquisition:** Maité Van Alboom, Liesbet Goubert, Piet Bracke, Bogdan Gabrys, Katarzyna Musial.

**Investigation:** Mingshan Jia, Maité Van Alboom.

**Methodology:** Mingshan Jia, Maité Van Alboom, Bogdan Gabrys, Katarzyna Musial.

**Project administration:** Mingshan Jia, Liesbet Goubert, Piet Bracke, Bogdan Gabrys, Katarzyna Musial.

**Resources:** Bogdan Gabrys, Katarzyna Musial.

**Software:** Mingshan Jia.

**Supervision:** Liesbet Goubert, Piet Bracke, Bogdan Gabrys, Katarzyna Musial.

**Validation:** Mingshan Jia.

**Visualization:** Mingshan Jia.

**Writing – original draft:** Mingshan Jia.

**Writing – review & editing:** Maité Van Alboom, Liesbet Goubert, Piet Bracke, Bogdan Gabrys, Katarzyna Musial.

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
