## [Decision Letter · Decision Letter 0]

7 Jul 2022

PONE-D-22-15936Encoding edge type information in graphletsPLOS ONE

Dear Dr. Jia,

Thank you for submitting your manuscript to PLOS ONE. After careful consideration, we feel that it has merit but does not fully meet PLOS ONE’s publication criteria as it currently stands. Therefore, we invite you to submit a revised version of the manuscript that addresses the points raised during the review process.

ACADEMIC EDITOR: Please address all the concerns raised by the reviewers.

We look forward to receiving your revised manuscript.

Kind regards,

Lun Hu

Academic Editor

PLOS ONE

Journal Requirements:

- https://link.springer.com/chapter/10.1007/978-3-030-93409-5_43

In your revision ensure you cite all your sources (including your own works), and quote or rephrase any duplicated text outside the methods section. Further consideration is dependent on these concerns being addressed.

Reviewers' comments:

Reviewer's Responses to Questions

**Comments to the Author**

1. Is the manuscript technically sound, and do the data support the conclusions?

Reviewer #1: Partly

Reviewer #2: Yes

2. Has the statistical analysis been performed appropriately and rigorously? 

Reviewer #1: No

Reviewer #2: Yes

3. Have the authors made all data underlying the findings in their manuscript fully available?

Reviewer #1: Yes

Reviewer #2: No

4. Is the manuscript presented in an intelligible fashion and written in standard English?

Reviewer #1: Yes

Reviewer #2: Yes

5. Review Comments to the Author

Reviewer #1: ID: PONE-D-22-15936

Title: Encoding edge type information in graphlets

Summary:

In this work, a framework to embed edge type information in graphlets and generate a Typed-Edge Graphlets Degree Vector (TyE-GDV) is introduced.

The manuscript is interesting; however, the following comment should be addressed :

Abstract :

- - - - - - - - - - -

1 – The brief methodology should be given in the abstract .

2 - The obtained results should be included in the abstract. The results should be included in terms of improvement ratio between the proposed and existing works .

Introduction Section :

- - - - - - - - - - - - - - - - - - - - - -

3 – The structure of the manuscript need to be included at the end of this section .

4 – A related work section should be included in the manuscript .

Background and Preliminaries Section :

- - - - - - - - - - - - - - - - - - - - - - - - - - - - - - - - - - -

5 – This section is fine . no comments .

Typed-Edge Graphlet Degree Vector Section :

- - - - - - - - - - - - - - - - - - - - - - - - - - - - - - - - - - - - - -

6 – More justification is required in this section .

Typed-Edge Degree, Colored Graphlets and 162 Heterogeneous Graphlets Section :

- - - - - - - - - - - - - - - - - - - - - - - - - - - - - - - - - - - - - - - - - - - - - - - - - - - - - - - - - - - - - - - - - - - -

7 – This section is fine . No comments .

Experiments and Analysis Section :

- - - - - - - - - - - - - - - - - - - - - - - - - - - -

8 – For more readability, more visual examples are required

Conclusion Section :

- - - - - - - - - - - - - - - - - - - - - -

8 – The limitation of this work should be clearly included in the conclusion section . Also, future work need to be included in this section .

References :

- - - - - - - - - - - - - -

9 – References need to be updated from literature (2022) .

- - - - - - - - - - - - - - - - - - - - - - - - - - - - - - - - - - - - - - - - - - - - - - - - - - - - - - - - - - - - - - - - - - - - - - - - - - - - - - - - - - - - - - - - - - - - - - - - - - - - - - - - - - - - - - - - - - - - - - - - - - - - - - - - - - - - - - - - - - - - - - - - - - - - - - - - - - - - - - - - - - - - - - - - - - - - - - - -

Reviewer #2: The paper introduces Typed-Edge Graphlets Degree Vectors (TyE-GDV) as a tool to analyze egocentric networks with different edge types. The method is applied to pain patient data, both for analysis and for prediction of pain sensation from the patient's social network.

The paper is well written and presents interesting ideas and results. I have a few minor comments:

1. While the code is made available on GitHub, the data is not (at least not obviously to be found in the GitHub repository). For means of reproducibility it should be provided.

2. Algorithm 3: The variables t_o_6, t_o_9 etc. used in the algorithm are not defined, neither in the algorithm nor in the text.

3. Lines 135 and 137: It should read "from 0 to |T_e| - 1" and "from 0 to |O| - 1", respectively, to match the examples given within the same sentences.

4. Line 160f: It would be helpful to add a reference to Fig. 2, which shows the 7 orbits of TyE-EGDV.

5. Equations (1) and (2): The derivation of the formulae should be briefly explained, in particular since (1) is claimed to be a correction of a result previously published in [23].

6. Fig. 4: The figure could be drawn as a radar chart, which might be more familiar to the usual reader than a parallel coordinates plot. The difference between the pain grades with respect to orbit types might also be more obvious in a radar chart. For Fig. 5 parallel coordinate plots are probably more appropriate due to the large number of attributes (edge types).

7. References: Starting from ref. [17], page numbers are missing for most references. Also the formatting of reference entries varies, e.g., regarding the capitalization of journal titles. The whole reference section should be thoroughly proofread and amended where necessary.

6. PLOS authors have the option to publish the peer review history of their article (what does this mean?). If published, this will include your full peer review and any attached files.

Reviewer #1: No

Reviewer #2: **Yes: **Volker Ahlers

---

## [Author Response · Author response to Decision Letter 0]

22 Jul 2022

Our detailed reponse is in the attached rebuttal letter.

---

## [Decision Letter · Decision Letter 1]

2 Aug 2022

PONE-D-22-15936R1Encoding edge type information in graphletsPLOS ONE

Dear Dr. Jia,

Thank you for submitting your manuscript to PLOS ONE. After careful consideration, we feel that it has merit but does not fully meet PLOS ONE’s publication criteria as it currently stands. Therefore, we invite you to submit a revised version of the manuscript that addresses the points raised during the review process.

We look forward to receiving your revised manuscript.

Kind regards,

Lun Hu

Academic Editor

PLOS ONE

Journal Requirements:

Additional Editor Comments:

Although the quality of this work has been improved a lot after revision, one reviewer still has some minor concerns about the paper, please address them for publication.

Reviewers' comments:

Reviewer's Responses to Questions

**Comments to the Author**

1. If the authors have adequately addressed your comments raised in a previous round of review and you feel that this manuscript is now acceptable for publication, you may indicate that here to bypass the “Comments to the Author” section, enter your conflict of interest statement in the “Confidential to Editor” section, and submit your "Accept" recommendation.

Reviewer #1: (No Response)

Reviewer #2: All comments have been addressed

2. Is the manuscript technically sound, and do the data support the conclusions?

Reviewer #1: Yes

Reviewer #2: (No Response)

3. Has the statistical analysis been performed appropriately and rigorously? 

Reviewer #1: Yes

Reviewer #2: (No Response)

4. Have the authors made all data underlying the findings in their manuscript fully available?

Reviewer #1: Yes

Reviewer #2: (No Response)

5. Is the manuscript presented in an intelligible fashion and written in standard English?

Reviewer #1: Yes

Reviewer #2: (No Response)

6. Review Comments to the Author

Reviewer #1: ID: PONE-D-22-15936 R1

Title: Encoding edge type information in graphlets

Summary:

In this work, a framework to embed edge type information in graphlets and generate a Typed-Edge Graphlets Degree Vector (TyE-GDV) is introduced.

The authors have addressed the raised comments; however, one comment need to be addressed. Please check the comments below :

Abstract :

- - - - - - - - - - -

1 – The abstract is fine. No comments .

Introduction Section :

- - - - - - - - - - - - - - - - - - - - - -

2 – This section is fine. No comments .

Preliminaries Section :

- - - - - - - - - - - - - - - - - - - - - - - - - - - - - - - - - - -

3 – This section is fine . no comments .

Typed-Edge Graphlet Degree Vector Section :

- - - - - - - - - - - - - - - - - - - - - - - - - - - - - - - - - - - - - -

4 – This section is fine . no comments .

Typed-Edge Degree, Colored Graphlets and 162 Heterogeneous Graphlets Section :

- - - - - - - - - - - - - - - - - - - - - - - - - - - - - - - - - - - - - - - - - - - - - - - - - - - - - - - - - - - - - - - - - - - -

5 – This section is fine . No comments .

Experiments and Analysis Section :

- - - - - - - - - - - - - - - - - - - - - - - - - - - -

6 – This section is fine . No comments .

Related Work Section :

- - - - - - - - - - - - - - - - - - - - - - - - - - - -

7 – This section need to be moved after the Introduction Section .

Conclusion Section :

- - - - - - - - - - - - - - - - - - - - - -

8 – This section is fine . No comments .

References :

- - - - - - - - - - - - - -

9 – The references are fine.

- - - - - - - - - - - - - - - - - - - - - - - - - - - - - - - - - - - - - - - - - - - - - - - - - - - - - - - - - - - - - - - - - - - - - - - - - - - - - - - - - - - - - - - - - - - - - - - - - - - - - - - - - - - - - - - - - - - - - - - - - - - - - - - - - - - - - - - - - - - - - - - - - - - - - - - - - - - - - - - - - - - - - - - - - - - - - - - - - - - - - - - - - - - - - - - - - - - - - - - - - - - - - - - - - - - - - - - - - - - - - - - - - - - - - - - - - - - - - - - - - - - - - - - - - - - - - - - - - - - - - - - - - - - - - - - - - - - - - - - - - - - - - - - - - - - - - - - - - - - - - - - - - - - - - - - - - - - - - - - - - - - - - - - - - - - - - - - - - - - - - - - - - - - - - - - -

Reviewer #2: (No Response)

7. PLOS authors have the option to publish the peer review history of their article (what does this mean?). If published, this will include your full peer review and any attached files.

Reviewer #1: No

Reviewer #2: **Yes: **Volker Ahlers

---

## [Author Response · Author response to Decision Letter 1]

2 Aug 2022

Please find our detailed response in the attached rebuttal letter.

---

## [Decision Letter · Decision Letter 2]

15 Aug 2022

Encoding edge type information in graphlets

PONE-D-22-15936R2

Dear Dr. Jia,

We’re pleased to inform you that your manuscript has been judged scientifically suitable for publication and will be formally accepted for publication once it meets all outstanding technical requirements.

Kind regards,

Lun Hu

Academic Editor

PLOS ONE

Additional Editor Comments (optional):

All reviewers agreed that this work can now be accepted.

Reviewers' comments:

Reviewer's Responses to Questions

**Comments to the Author**

1. If the authors have adequately addressed your comments raised in a previous round of review and you feel that this manuscript is now acceptable for publication, you may indicate that here to bypass the “Comments to the Author” section, enter your conflict of interest statement in the “Confidential to Editor” section, and submit your "Accept" recommendation.

Reviewer #1: All comments have been addressed

Reviewer #2: All comments have been addressed

2. Is the manuscript technically sound, and do the data support the conclusions?

Reviewer #1: Yes

Reviewer #2: (No Response)

3. Has the statistical analysis been performed appropriately and rigorously? 

Reviewer #1: Yes

Reviewer #2: (No Response)

4. Have the authors made all data underlying the findings in their manuscript fully available?

Reviewer #1: Yes

Reviewer #2: (No Response)

5. Is the manuscript presented in an intelligible fashion and written in standard English?

Reviewer #1: Yes

Reviewer #2: (No Response)

6. Review Comments to the Author

Reviewer #1: ID: PONE-D-22-15936 R2

Title: Encoding edge type information in graphlets

Summary:

In this work, a framework to embed edge type information in graphlets and generate a Typed-Edge Graphlets Degree Vector (TyE-GDV) is introduced.

The authors have addressed all the raised comments.

Abstract :

- - - - - - - - - - -

1 – The abstract is fine. No comments .

Introduction Section :

- - - - - - - - - - - - - - - - - - - - - -

2 – This section is fine. No comments .

Related Work Section :

- - - - - - - - - - - - - - - - - - - - - - - - - - - -

3 – This section is fine. No comments .

Preliminaries Section :

- - - - - - - - - - - - - - - - - - - - - - - - - - - - - - - - - - -

4 – This section is fine . no comments .

Typed-Edge Graphlet Degree Vector Section :

- - - - - - - - - - - - - - - - - - - - - - - - - - - - - - - - - - - - - -

5 – This section is fine . no comments .

Typed-Edge Degree, Colored Graphlets and 162 Heterogeneous Graphlets Section :

- - - - - - - - - - - - - - - - - - - - - - - - - - - - - - - - - - - - - - - - - - - - - - - - - - - - - - - - - - - - - - - - - - - -

6 – This section is fine . No comments .

Experiments and Analysis Section :

- - - - - - - - - - - - - - - - - - - - - - - - - - - -

7 – This section is fine . No comments .

Conclusion Section :

- - - - - - - - - - - - - - - - - - - - - -

8 – This section is fine . No comments .

References :

- - - - - - - - - - - - - -

9 – The references are fine.

- - - - - - - - - - - - - - - - - - - - - - - - - - - - - - - - - - - - - - - - - - - - - - - - - - - - - - - - - - - - - - - - - - - - - - - - - - - - - - - - - - - - - - - - - - - - - - - - - - - - - - - - - - - - - - - - - - - - - - - - - - - - - - - - - - - - - - - - - - - - - - - - - - - - - - - - - - - - - - - - - - - - - - - - - - - - - - - - - - - - - - - - - - - - - - - - - - - - - - - - - - - - - - - - - - - - - - - - - - - - - - - - - - - - - - - - - - - - - - - - - - - - - - - - - - - - - - - - - - - - - - - - - - - - - - - - - - - - - - - - - - - - - - - - - - - - - - - - - - - - - - - - - - - - - - - - - - - - - - - - - - - - - - - - - - - - - - - - - - - - - - - - - - - - - - - - - - - - - - - - - - - - - - - - - - - - - - - - - - - - - - - - - - - - - - - - - - - - - - - - - - - - - - - - - - - - - - - - - - - - - - - - - - - - - - - - - - - - - - - - - - - - - - - - - - - - - - - - - - - - - - - - - - - - - - - - - - - - - - - - - - - - - - - - - - - - - - - - - - - - - - - - - - - - - - - - - - - - - - - - - - - - - - - - - - - - - - - - - - - - - - - - - - - - - - - - - - - - - - - - - - - - - - - - - - - - - - - - - - - - - - - - - - - - - - - - - - - - - - - - - - - - - - - - - - - - - - - - - - - - - - - - - - - - - - - - - - - - - - - - - - - - - - - - - - - - - - - - - - - - - - - - - - - - - - - - - - - - - - - - - - - - - - - - - - - - - - - - - - - - - - - -

Reviewer #2: (No Response)

7. PLOS authors have the option to publish the peer review history of their article (what does this mean?). If published, this will include your full peer review and any attached files.

Reviewer #1: No

Reviewer #2: **Yes: **Volker Ahlers

---

## [Editor Report · Acceptance letter]

17 Aug 2022

PONE-D-22-15936R2 

Encoding edge type information in graphlets 

Dear Dr. Jia:

I'm pleased to inform you that your manuscript has been deemed suitable for publication in PLOS ONE. Congratulations! Your manuscript is now with our production department. 

Kind regards, 

on behalf of

Dr. Lun Hu 

Academic Editor

PLOS ONE